# C-terminal truncation modulates α-Synuclein's cytotoxicity and aggregation by promoting the interactions with membrane and chaperone

Cai Zhang[1,2,5], Yunshan Pei[1,2,5], Zeting Zhang [1,3 ✉], Lingling Xu[1], Xiaoli Liu[1], Ling Jiang[1,3], Gary J. Pielak [4], Xin Zhou[1,3], Maili Liu[1,2,3] & Conggang Li [1,2,3 ✉]

α-Synuclein (α-syn) is the main protein component of Lewy bodies, the major pathological hallmarks of Parkinson's disease (PD). C-terminally truncated α-syn is found in the brain of PD patients, reduces cell viability and tends to form fibrils. Nevertheless, little is known about the mechanisms underlying the role of C-terminal truncation on the cytotoxicity and aggregation of α-syn. Here, we use nuclear magnetic resonance spectroscopy to show that the truncation alters α-syn conformation, resulting in an attractive interaction of the N-terminus with membranes and molecular chaperone, protein disulfide isomerase (PDI). The truncated protein is more toxic to mitochondria than full-length protein and diminishes the effect of PDI on α-syn fibrillation. Our findings reveal a modulatory role for the C-terminus in the cytotoxicity and aggregation of α-syn by interfering with the N-terminus binding to membranes and chaperone, and provide a molecular basis for the pathological role of C-terminal truncation in PD pathogenesis.

[1] Key Laboratory of Magnetic Resonance in Biological Systems, State Key Laboratory of Magnetic Resonance and Atomic and Molecular Physics, National Center for Magnetic Resonance in Wuhan, Wuhan Institute of Physics and Mathematics, Innovation Academy for Precision Measurement Science and Technology, Chinese Academy of Sciences, 430071 Wuhan, China. [2] Graduate University of Chinese Academy of Science, 100049 Beijing, China. [3] Wuhan National Laboratory for Optoelectronics, Huazhong University of Science and Technology, 430071 Wuhan, China. [4] Department of Chemistry, Department of Biochemistry and Biophysics, Lineberger Comprehensive Cancer Center, Integrative Program for Biological and Genome Sciences, University of North Carolina, Chapel Hill, NC 27599, USA. [5] These authors contributed equally: Cai Zhang, Yunshan Pei. ✉email: zhangzeting@apm.ac.cn; conggangli@apm.ac.cn

The soluble, disordered, monomeric protein, α-synuclein (α-syn) is abundant in presynaptic neurons and involved in neuronal function[1]. Insoluble aggregates of α-syn in Lewy bodies (LBs) are the hallmark of Parkinson's disease (PD)[2,3]. α-Syn's primary structure comprises three regions: the basic N-terminal domain (residues 1–60), which adopts an α-helix upon binding to negatively charged membranes[4–6]; the hydrophobic NAC domain (residues 61–95), which is prone to aggregation and form fibrils[7–9]; and the flexible C-terminal domain (residues 96–140), which contains many negatively charged residues and interacts with cations and polyamines[10,11].

Multiple forms of C-terminally truncated α-syn (CT-α-syn) are detected in normal and PD brains[12–16], and widely investigated due to their remarkable ability to aggregate and transform into pathologic fibrils. CT-α-syn accelerates formation of oligomers and fibrils compared to the full-length protein (FL-α-syn) in vitro[17–27]. When co-expressed with FL-α-syn, CT-α-syn promotes the pathological accumulation of FL-α-syn[28,29]. Deleting C-terminal residues accelerates aggregation up to residues 85-90, where the NAC domain begins. Further truncation decreases aggregation propensity because the NAC region forms the core of amyloid fibrils[7,18,30]. Cells expressing CT-α-syn are more vulnerable to oxidative stress and CT-α-syn is more toxic than FL-α-syn[21,31–33]. Transgenic mice expressing C-terminal truncated species manifest PD-like symptoms[34]. These observations suggest a role for CT-α-syn in PD, but the causes are largely unknown.

Based on this variety of cytotoxicity, plenty of attention has been paid to the role of the C-terminus in α-syn pathology, such as C-terminal-engaged interactions and the effect of the C-terminus on fibrillation. The native compact conformation of monomeric α-syn is driven by the long-range transient intramolecular interactions between the N- and C-termini, release of which accelerates aggregation[35,36]. α-Syn's N-terminus is responsible for the fibril core, but also interacts with membranes and chaperones, which modulates fibrillation[37,38]. For instance, the N-terminal domain regulates fusion of synaptic vesicles and subsequent neurotransmitter release, which are critical for normal brain function[39,40]. Molecular chaperones, such as protein disulfide isomerase (PDI) and Hsp70, bind to the N-terminal domain, dissolve and degrade accumulated misfolded α-syn, highlighting the role of chaperones in preventing α-syn aggregation and regulating its physiological function[41–43]. Yet it is unknown whether long-range intramolecular interactions between the N- and C-terminus affect the association of the N-terminus with other molecules and the implications in pathological toxicity. Does C-terminal truncation free the N-terminal domain to interact with membranes and chaperones, thus affecting α-syn functionality?

Here, we investigate the effects of C-terminal truncation on the interactions of the N-terminus with membranes and molecular chaperones using FL- and CT-α-syn (residues 1-99). Nuclear magnetic resonance spectroscopy (NMR) and coflotation data reveal a stronger interaction of CT-α-syn with membranes and with PDI, a major endoplasmic reticulum chaperone[44]. The interaction may arise from a more extended conformation of CT-α-syn, as opposed to the compact conformation of FL-α-syn[45]. The functional assay reveals severe mitochondrial membrane disruption by CT-α-syn compared to FL-α-syn in both neuronal cells and isolated mitochondria. Dramatically differential inhibitory effects of PDI on FL-α-syn and CT-α-syn fibrillation are also observed. Our results suggest that the C-terminus behaves like a 'guard' regulating α-syn function. In addition to homogeneous intermolecular interplays associated with self-assembly, it affects N-terminal interactions with other molecules. The absence of this 'guard' exposes the N-terminus, resulting in stronger interaction with membranes and chaperone, mitochondrial dysfunction and diminishes the inhibitory effect of chaperone, all of which could affect the etiology of PD.

## Results

**C-terminal truncation causes stronger interactions with membranes and PDI.** Under physiological conditions, about 15% of α-syn exists as membrane-bound form[46]. Several studies highlight a specific interaction of α-syn with the acid phospholipid cardiolipin (CL), a lipid unique to mitochondria, which is postulated to be associated with PD[47–50]. α-Syn forms a N-terminal α-helix upon binding to negatively charged membranes, without involvement of the C-terminus[4], but it is not known if deleting the C-terminus affects binding. We thus prepared mixtures comprising palmitoyl-2-oleoyl-sn-glycero-3-phosphocholine (POPC)/CL (7:3 mole ratio) to mimic mitochondrial membranes and POPC/1-palmitoly-2-oleoyl-sn-glycero-3-phosphate (POPA) (1:1 mole ratio) lipid mixture to obtain highly anionic liposomes. Binding of FL- and CT-α-syn (residues1-99) was analyzed by coflotation, a common method for studying protein-lipid interaction. Liposomes, and liposomes-associated proteins float to the top of the gradient and unbound proteins to the bottom (Fig. 1b). The fractions from top to bottom were collected and quantified by sodium dodecyl sulfate polyacrylamide gel electrophoresis (SDS-PAGE) (Fig. 1c and Supplementary Fig. 1a). For both synthetic mitochondrial membranes and 50% POPA anionic membranes, ~80% of the CT-α-syn and 50–60% of FL-α-syn binds (Fig. 1d, Supplementary Fig. 1b and Supplementary Data 1), indicating stronger association of CT-α-syn. Considering packing defects possibly promoting the interaction between CT-α-syn and membrane due to the small headgroup of POPA and CL, we employed 30% POPS (1-palmitoyl-2-oleoyl-sn-glycero-3-phospho-l-serine), a similar anionic lipid but with a larger headgroup. Similarly, the population of membrane-bound CT-α-syn is larger than that of membrane-bound FL-α-syn (50% versus 25%, Supplementary Fig. 1a, b), suggesting that CT-α-syn interacts more strongly with membranes.

To further confirm our findings, we acquired $^1$H-$^{15}$N heteronuclear single quantum coherence (HSQC) spectra of uniformly $^{15}$N-enriched FL- and CT-α-syn in the presence of POPC/CL (7:3 mole ratio) liposomes, POPC/POPA (1:1 mole ratio) liposomes, POPC/POPS (7:3 mole ratio) liposomes, respectively. NMR provides binding information at the residue level because resonances from strongly bound residues show reduced intensities[51]. Analysis of spectra acquired in the absence and presence of liposomes (Fig. 1e and Supplementary Fig. 1c, d) indicate that, consistent with previous results, the N-terminus dominates binding[5]. Data for CT-α-syn show that signals are more attenuated in the presence of liposomes containing CL, POPA or POPS compared to FL-α-syn, indicating a stronger binding of CT-α-syn to membranes than FL-α-syn, consistent with the coflotation data.

Molecular chaperones are another important binding partner of the N-terminal region of α-syn in mammalian cells[38]. PDI interacts with the hydrophobic region of the N-terminal region, inhibiting aggregation[43,52,53]. To investigate the impact of the C-terminal truncation on the interaction with PDI, we acquired $^1$H-$^{15}$N HSQC spectra of uniformly $^{15}$N-enriched FL- and CT-α-syn in the presence of PDI at a mole ratio of 2:1, which enabled us to assess intermolecular interactions at the residue level via chemical shift perturbations (CSPs) and intensity changes. PDI binds the first 20 residues of FL-α-syn (segment 1) and 10 residues near Tyr39 (segment 2) (Fig. 2a, b). Both sites are located at the N-terminus, consistent with other reports[38,43]. Although the attenuation patterns of CT- and FL-α-syn in PDI are similar,

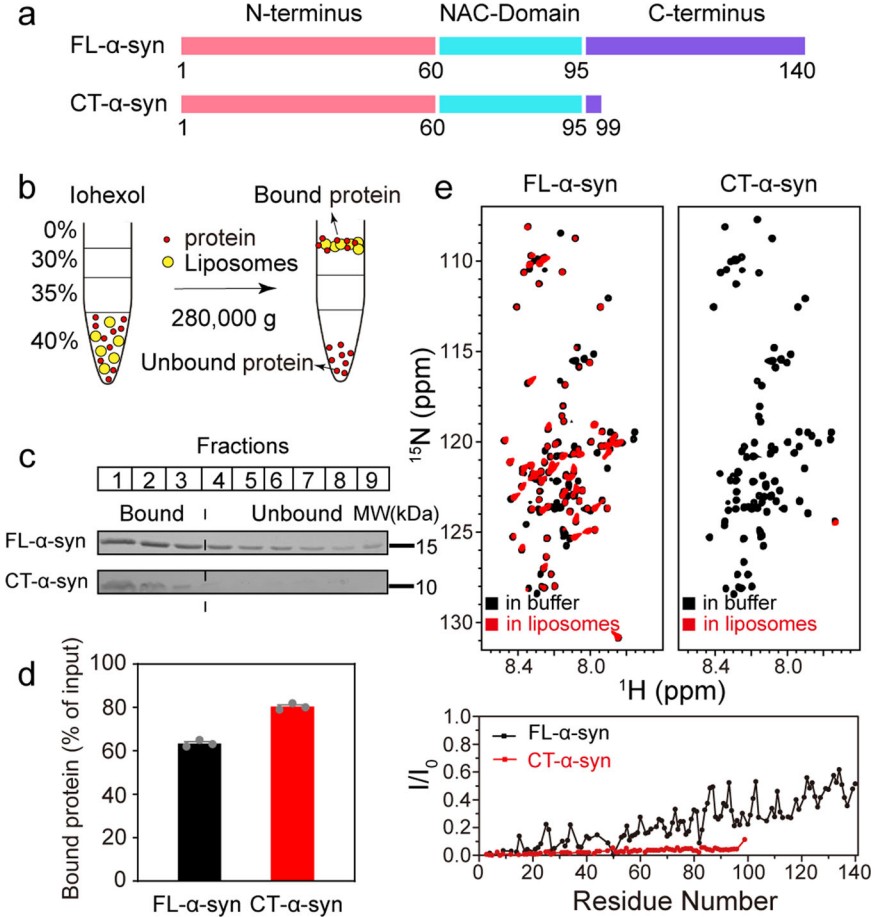

**Fig. 1 Stronger interaction of CT-α-syn with membrane. a** Primary structure schematic of FL- and CT-α-syn. **b** Cartoon of coflotation assay. **c** SDS-PAGE of fractions post centrifugation using POPC/CL (7:3 mole ratio) liposomes. FL- and CT-α-syn were run on SDS-PAGE independently. **d** The proportion of lipid-bound protein from SDS-PAGE data. Uncertainties are the standard error of the mean from three independent trials. **e** Overlaid $^1$H-$^{15}$N HSQC spectra of 0.12 mM $^{15}$N-enriched FL- and CT-α-syn in the absence (black) and presence (red) of 12 mM POPC/CL (7:3 mole ratio) liposomes and residue-resolved attenuation ($I/I_0$) of FL- (black circles) and CT-α-syn (red circles) upon adding liposomes. Values <1.0 indicate interactions.

there are differences in chemical shift and intensity (Fig. 2c), suggesting the constructs have similar PDI-binding sites but different affinities. As most signals of α-syn in segment 1 and 2 disappear when the mole ratio of PDI to α-syn is greater than 1:1, we quantied the dissociation constants ($K_D$) of FL- and CT-α-syn by fitting the average CSPs values of residues within segment 1 and segment 2 as a function of the concentration of PDI (Fig. 2d and Supplementary Data 2) up to 0.3 mM. CSP values of FL- and CT-α-syn at 0.6 mM PDI were predicted by fitting the curve. The fitting results show that segment 2 of FL- and CT-α-syn have similar $K_D$ values (~400 μM), whereas segment 1 of CT-α-syn has a $K_D$ of ~40 μM, which is much smaller than that of FL-α-syn (~700 μM).

**C-terminal truncated α-syn exhibits a more extended conformation.** Given the difference in membrane binding between FL- and CT-α-syn, we examined the constructs using paramagnetic relaxation enhancement (PRE) to determine whether deleting the C-terminus alters the ensemble of conformations. We prepared spin-labeled proteins by incorporating 1-oxyl-2,2,5,5-tetramethyl-delta3-pyrroline-3-methyl (MTSL), a paramagnetic spin label, into single cysteine variants (A90C) of $^{15}$N-enriched FL- and CT-α-syn. Cross-peaks arising from residues close to the label broaden due to enhanced transverse relaxation. As reported[35], intramolecular interactions between the N- and C-terminal regions lead to broadening proximal to the spin label

and in residues within the N-terminal region, neighboring Tyr39 (Fig. 3a), while the N-terminal residues of CT-α-syn show weak PRE effects (Fig. 3b). These results indicate that the interactions between the N- and C-terminal regions cause the PRE effects observed in FL-α-syn, and similar intramolecular long-range interactions are absent in CT-α-syn. We conclude that CT-α-syn exhibits a more extended conformation.

**Effects of C-terminus truncation on physiological and pathological functions.** α-Syn localizes to mitochondria in the human brain, and the association of α-syn with the mitochondrial membrane is related to mitochondrial dysfunction[54–56]. To better understand the membrane-related physiological differences between FL- and CT-α-syn, we evaluated the mitochondrial potential in living SK-N-SH cells using the dye, 5, 5′, 6, 6′-tetrachloro-1, 1′, 3, 3′-tetraethylbenzimidazolocarbocyanine iodide (JC-1). In normal cells, JC-1 aggregates in the matrix due to the electrochemical potential gradient resulting in the red fluorescence. Apoptotic events cause monomeric JC-1 to disperse throughout the cell resulting in green fluorescence. Images of SK-N-SH cells were acquired after treatment with 50 μM FL- or CT-α-syn for 5 h. Mitochondria in untreated SK-N-SH cells stain bright red, but red fluorescence decreased markedly in CT-α-syn-incubated cells concomitant with an increase in green fluorescence, indicating collapse of the mitochondrial membrane potential (Fig. 4a). The green/red fluorescence intensity ratio showed no significant difference between FL-α-syn-treated and control cells

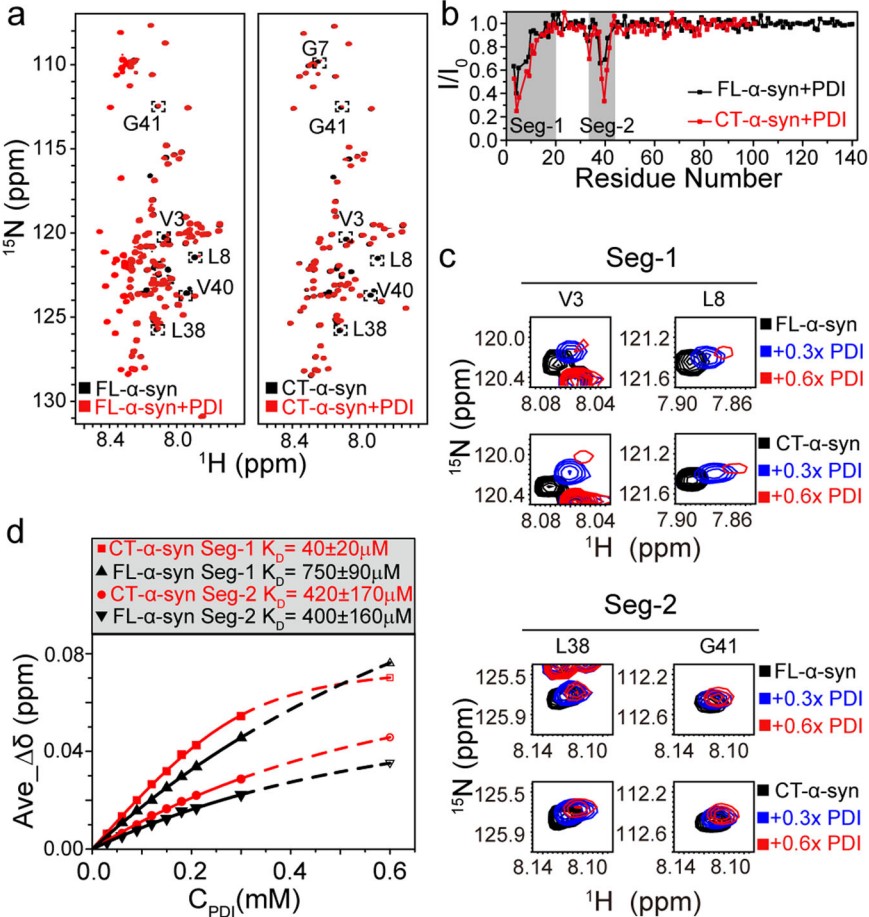

**Fig. 2 Stronger interaction of CT-α-syn with PDI. a** Overlay of $^1H$-$^{15}N$ HSQC spectra of 0.3 mM $^{15}N$-enriched FL- and CT-α-syn in the absence (black) and presence (red) of 0.15 mM PDI. Perturbed residues are surrounded by a dotted box. **b** Residue-resolved signal attenuation ($I/I_0$) of 0.3 mM FL- (black circles) and CT-α-syn (red circles) upon adding 0.15 mM PDI. Values <1.0 indicate interactions. Binding regions are colored gray. **c** Representative $^1H$-$^{15}N$ HSQC cross-peaks within segments 1 and 2 of 0.3 mM FL- and CT-α-syn titrated with PDI. **d** Average CSPs of distinguishable residues within segment 1 (V3, M5, G7, L8, S9, K12, V15, A17, A18, A19, and E20) and 2 (T33, V37, L38, V40, G41, and T44) of FL- and CT-α-syn as a function of the PDI concentration. $K_D$ values from curve fitting are shown. CSP values of FL- and CT-α-syn at 0.6 mM PDI were predicted by fitting the curve due to signal attenuation from segments 1 and 2 at higher (>0.3 mM) PDI concentration.

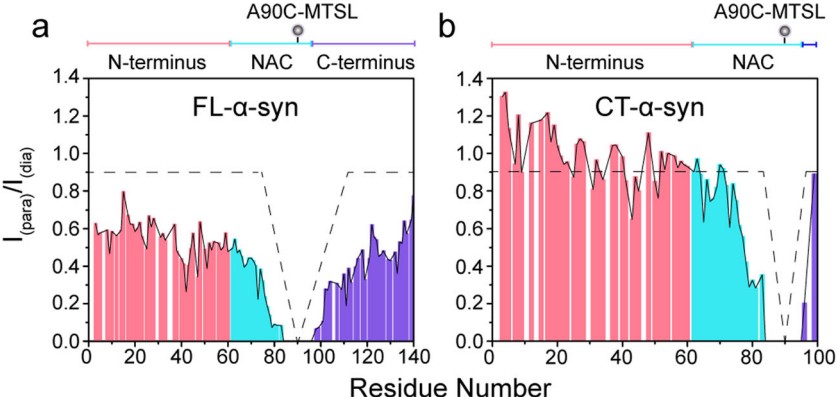

**Fig. 3 A more extended conformation for CT-α-syn. a, b** Intramolecular PRE intensity ratios of backbone amides in FL- (**a**) and CT-α-syn variants (**b**) possessing a MTSL spin label at A90C. Pink, blue, and burgundy identify the N-terminal, NAC and the C-terminal domain, respectively. Schematic diagram of sequence domain is shown above each plot. Circles indicate locations of the label.

(Fig. 4b and Supplementary Data 3). Thus, the C-terminal truncation increases mitochondrial damage compared to the negative control and FL-α-syn. We excluded the possibility that CT-α-syn aggregated into oligomers or fibrils during 5 h incubation with SK-N-SH cells

(Supplementary Fig. 2), thus the cytotoxicity of CT-α-syn may be mainly caused by enhanced monomer-membrane interactions.

To further confirm the functional consequences of CT-α-syn on mitochondria, we isolated mitochondria from SK-N-SH cells.

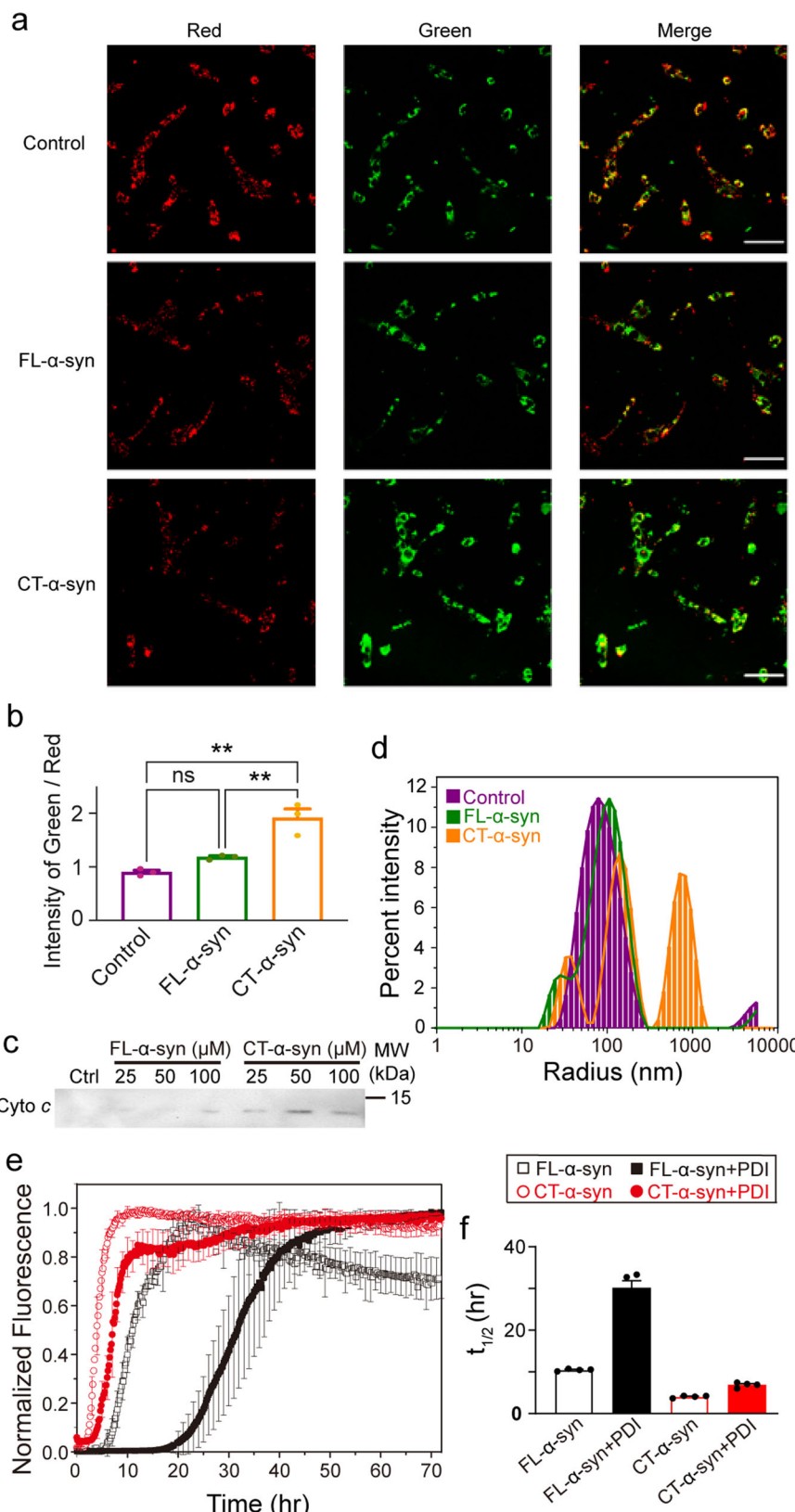

Given the dissipated membrane potential, we anticipated mitochondrial membrane permeabilization, allowing cytochrome *c* (cyto *c*) release[57,58]. We assessed release of cyto *c* after treatment of mitochondria with various concentrations of FL- and CT-α-syn for 1 h at 37 °C (Fig. 4c) via Western blotting. Enhanced release relative to control mitochondria was detected with CT-α-

syn. We did not observe a significant release of cyto *c* compared to controls with FL-α-syn. This result lends support to the idea that CT-α-syn disrupts mitochondrial membrane, leading to cyto *c* release.

To understand the influence of C-terminal truncation on membrane interactions, we used dynamic light scattering to

**Fig. 4 Physiological and pathological consequences of CT-α-syn. a** Representative confocal images of JC-1-stained SK-N-SH cells after 5 h treatment with 50 μM FL-α-syn and CT-α-syn at 37 °C. Scale bar = 2 μm. **b** Integrated fluorescence intensities of five random visual fields were analyzed with Image J software using the mean ratio of green to red fluorescence to indicate mitochondrial damage. Uncertainties are the standard error of the mean from three biologically independent experiments. Statistical analysis were performed by one-way ANOVA for multiple comparisons. $p < 0.05$ was considered statistically significant difference (*$p < 0.05$; **$p < 0.01$; ns, not significant). **c** Cytochrome c levels were assessed by Western blotting after incubating isolated mitochondria with increasing concentrations of FL- and CT-α-syn at 37 °C for 1 h. Unincubated mitochondria served as positive controls. **d** Dynamic light scattering-measured particle size of POPC/CL (7:3 mole ratio) liposomes alone (purple line), or after incubation with FL- (green line) or CT-α-syn (orange line) for 30 min. **e** Aggregation kinetics of 0.3 mM FL- and CT-α-syn in the absence and presence of equimolar PDI monitored by thioflavin T fluorescence. **f** Half-time ($t_{1/2}$) of aggregation. Uncertainties are the standard error of the mean from four independent experiments.

### Table 1 Aggregation kinetics.

| Protein | $t_{lag}$ (h)[a] | $k$ [b] | $t_{1/2}$ (h)[c] |
|---|---|---|---|
| FL-α-syn | 7.4 ± 0.4 | 0.65 ± 0.03 | 10.5 ± 0.3 |
| CT-α-syn | 2.1 ± 0.3 | 1.1 ± 0.1 | 4.0 ± 0.3 |
| Y39E-α-syn | ND[d] | ND[d] | ND[d] |
| FL-α-syn +PDI | 23 ± 2 | 0.28 ± 0.05 | 30 ± 3 |
| CT-α-syn +PDI | 4.0 ± 0.3 | 0.7 ± 0.1 | 6.9 ± 0.6 |
| Y39E-α-syn+PDI | 36 ± 2 | 0.21 ± 0.03 | 46 ±1 |

[a]Lag time.
[b]Elongation rate constant.
[c]Half-time of aggregation.
[d]Plateau was not reached.

monitor the formation of POPC/CL liposome clusters in the presence of FL- and CT-α-syn. FL-α-syn barely clusters liposomes while CT-α-syn dramatically increases particle size (Fig. 4d). The results agree with our NMR data showing that CT-α-syn interacts more strongly with liposomes than FL-α-syn. Hence, the ability of CT-α-syn to cluster liposomes arises from a stronger membrane interaction, which could disrupt membranes.

To investigate the effect of chaperone binding on CT-α-syn aggregation, we used Thioflavin T (ThT) fluorescence to monitor the rate of CT-α-syn aggregation in the absence or presence of an equal molar ratio of PDI and compared the rate to that of FL-α-syn. The ß-sheet structure of amyloid fibrils binds ThT dye enhancing fluorescence[59]. Figure 4e shows a typical nucleation-aggregation profile, beginning with a lag phase, followed by an elongation phase, and ending in a plateau. In the absence of PDI, truncation of the C-terminus accelerates CT-α-syn aggregation with a decreased lag time compared to FL-α-syn (Table 1, Fig. 4e and Supplementary Data 3). This result points to an auto-inhibitory role for the C-terminus on fibrillation, consistent with the previous studies[20,23,60]. As expected, the presence of PDI inhibits FL-α-syn aggregation[52,61,62], but inhibits CT-α-syn aggregation to a somewhat smaller extent, as reflected by an over 3-fold increase in lag time for FL-α-syn whereas less than 2-fold for CT-α-syn. Besides, half-time of aggregation ($t_{1/2}$) clearly shows that PDI exerted a less inhibitory effect on CT-α-syn aggregation (Fig. 4f and Supplementary Data 3), though segment 1 of CT-α-syn interacting more strongly with PDI.

Segment 2, comprising residues 36–42, is reported to be the 'master controller' of α-syn aggregation at neutral pH[63]. We hypothesized that the stronger interaction of PDI with segment 1 of CT-α-syn explains the diminished inhibitory effect on aggregation. To test this idea, we changed Tyr39 to Glu (Y39E) in segment 2 to weaken its affinity for PDI. As expected, the mutation abolishes the interaction of segment 2 with PDI whereas segment 1 still bound tightly (Supplementary Fig. 3a–c). While $t_{1/2}$ of the variant in the absence of PDI was too long to measure, the ThT fluorescence assay indicates an increased aggregation rate in the presence of PDI, with $t_{1/2}$ of about 45 h (Table 1, Supplementary Fig. 3d, e), suggesting that segment 2 is a controller and segment 1 acts as an accelerator of fibrillation upon binding to PDI. These results indicate that the interaction of PDI with segment 1 is enhanced in CT-α-syn, resulting in a diminished inhibition effect of PDI on fibrillation.

### Discussion

Misfolded, oligomeric, and fibrillar α-syn contribute to neuronal toxicity and progression of PD[64–66]. The proposed transient long-range interaction between the N-terminus and acidic tail of α-syn may inhibit fibril formation[67]. Releasing these contacts destabilizes the natively unfolded monomers potentiating aggregation[35,68]. For instance, $Ca^{2+}$, $Cu^{2+}$ and polyamine binding alters α-syn's long-range interaction exposing the amyloidogenic NAC region, resulting in enhanced fibrillation[60]. Here, we observe a conformational change for C-terminally truncated α-syn. The results suggest that deleting the C-terminus disrupts the intramolecular long-range interaction between the N- and C-terminus such that CT-α-syn adopts a relatively extended conformation in which the N-terminus is more exposed (Fig. 3b), suggesting an auto-chaperoning role of the C-terminus by shielding the N-terminus of α-syn, in agreement with previous reports[20,69–72].

Our finding that truncating the C-terminal region facilitates membrane binding contradicts a report purporting to show that the deletion has no effect[73]. However, our NMR data (Fig. 1e and Supplementary Fig. 1c, d) show attenuated signals of CT-α-syn in the presence of liposomes compared to FL-α-syn, supporting our coflotation data (Fig. 1c, d and Supplementary Fig. 1a, b) indicating more membrane-bound CT-α-syn compared to FL-α-syn. The dynamic light scattering data (Fig. 4d) also demonstrate stronger interaction of CT-α-syn with membrane. α-Syn binds membranes via electrostatic interactions involving the positively charged N-terminus with the negatively charged acid headgroups of phospholipid, and hydrophobic interactions between hydrophobic residues and fatty acyl chain of the phospholipid[74]. Hence, exposing the N-terminus strengthens the binding to liposomes.

α-Syn directly associates with mitochondrial membrane, and membrane interaction is a common cause of membrane disruption by the intrinsically disordered proteins[48,75]. After establishing the stronger interaction of CT-α-syn with artificial mitochondrial membranes, we turned to the physiological effect on mitochondrial membrane potential in living SK-N-SH cells. Using JC-1 staining, the data (Fig. 4a, b) show CT-α-syn collapses the potential, which coincides with cyto c release (Fig. 4c). By contrast, FL-α-syn does not impact cyto c release or membrane potential. α-Syn causes mitochondrial fragmentation via direct interaction with mitochondrial membranes, and these fragmented mitochondria are present in neurons and LBs along with α-syn deposits, including truncated forms[76–79]. The stronger interaction of CT-α-syn with mitochondrial membranes sheds light on the basis of toxicity and mitochondrial dysfunction induced by CT-α-syn.

Molecular chaperones help dissolve α-syn aggregates and prevent formation of pathological fibrils. For instance, Hsp70, a mammalian chaperone, breaks α-syn fibrils into monomers or

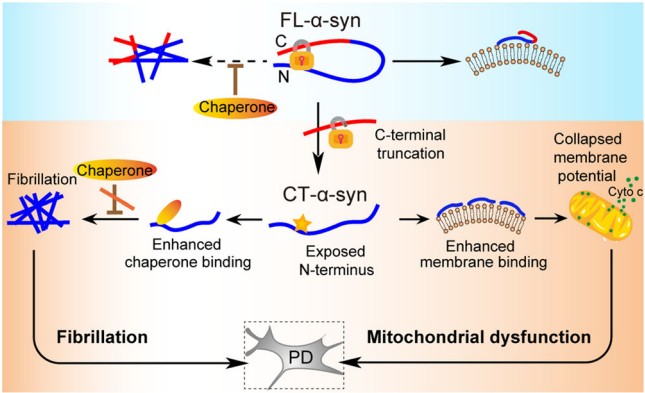

**Fig. 5 Possible mechanism of CT-α-syn involvement in the pathogenesis of PD.** The interaction between FL-α-syn and membranes is critical for neurotransmitter release, and fibrillation of FL-α-syn is inhibited by chaperones. When the C-terminus is truncated, the interaction with membranes and chaperones becomes stronger, leading to diminished inhibition of fibrillation and mitochondrial damage that is likely to elicit a truncation cycle contributing to pathogenesis.

oligomers[41,42]. The common chaperone binding region of α-syn is in the N-terminus. Our data showing that PDI binds two segments in the N-terminus (Fig. 2a–c) are consistent with the result of Yagi-Utsumi et al.[43] In addition, we find that segment 1 in CT-α-syn binds more tightly to PDI, compared to FL-α-syn, while segment 2 has a similar affinity (Fig. 2d). Using in vitro fibrillation assay, we find that the stronger interaction of segment 1 diminishes the inhibitory effect of PDI on CT-α-syn aggregation (Table 1, Fig. 4e, f). Upon decreasing the affinity of segment 2 for PDI and retaining the segment 1 by mutating Tyr39 to Glu (Y39E, Supplementary Fig. 3a–c), we observed faster aggregation in the presence than in the absence of PDI (Table 1, Supplementary Fig. 3d, e), further supporting the role of segment 1 as an accelerator of fibrillation when binding to PDI. Our results reinforce the importance of residues near Tyr39 in manipulating aggregation, reveal the differential effect of segment 1 and 2 on the fibrillation upon binding to PDI, and may provide a clue for chaperone-based design of fibrillation inhibitors. Nevertheless, the C-terminal truncation frees the N-terminus also enhancing the association with molecular chaperone, which is eventually reflected in the weakened inhibitory effect on CT-α-syn fibrillation.

Our results, along with those others[35,36], demonstrate the presence of the long-range interactions between the N- and C-terminal regions which are disrupted by the C-terminal truncation leaving the N-terminus largely exposed. In this work, we find that the C-terminal region of α-syn balances the interactions of the N-terminus with the membranes and chaperones. In view of this, we propose a mechanism for how CT-α-syn might be involved in PD pathogenesis(Fig. 5). The long-range interactions between the N- and C-terminal regions keep α-syn in its compact conformation, while the more extended conformation of CT-α-syn favors the heterogeneous intermolecular interactions of α-syn, such as with membrane and chaperone. Chaperones help cells handle misfolded proteins but when the C-terminus is truncated, segment 1 in the exposed N-terminus has an enhanced interaction with PDI resulting in partly uncontrolled fibrillation. Moreover, the stronger interaction between the CT-α-syn and mitochondrial membrane collapses the mitochondrial membrane potential and enhances membrane permeability, causing cyto *c* release. Based on previous studies and our results, we proposed that the mitochondrial dysfunction likely activates the apoptosome and the NLRP3 inflammasome[80–85], which may lead to

more truncation of FL-α-syn, forming a negative-feedback loop that amplifies truncation and elicits cell death[86–88]. Our observations highlight the guard role of the C-terminus in the cytotoxicity and aggregation of α-syn and show the importance of maintaining intact α-syn in cells. We propose that reducing C-terminal truncation is a plausible way to prevent the onset or development of PD.

The C-terminal truncation releases the long-range interactions between the N- and C-terminal domains of α-syn. CT-α-syn exhibits a more extended conformation, resulting in stronger interactions with membranes and molecular chaperones. The truncation further causes mitochondrial damage and accelerates fibrillation, both are key aspects of pathogenesis. We conclude that the C-terminus plays an auto-inhibitory role in FL-α-syn, shielding the N-terminus from heterogeneous intermolecular interactions, and acting as a 'guard', fending off cytotoxicity and aggregation of α-syn by maintaining moderate α-syn-membrane interactions and inhibitory effect of chaperones on fibrillation. We speculate that preventing truncation is a strategy for preventing the onset or development of PD and synucleinopathies[26,27,33,89].

## Methods

**Protein expression and purification**. α-Syn mutants containing a deletion of the C-terminus (amino acids 1-99), single cysteine mutants and single glutamic acid mutant were generated by using site-directed mutagenesis. Natural abundance and [15]N-enriched α-syn and its variants were expressed in *Escherichia coli* BL21 (DE3) cells in either LB media or M9 minimal medium supplemented with [15]NH₄Cl. The cells were grown at 37 °C to an optical density at 600 nm (OD₆₀₀) of 0.6 and induced with 1 mM IPTG for 4 h. Cells were harvested by centrifugation at 6300 *g* for 10 min at 4 °C. FL-α-syn was purified as described[90]. Cells expressing FL-α-syn or FL-A90C were homogenized by a high-pressure homogenizer (ATS Engineering) prior to boiling the cell lysates for 20 min, making most cellular proteins precipitated. Then streptomycin sulfate (10 mg/mL) was added to supernatant and was rotated at 4 °C for 30 min followed by the centrifugation at 14,000*g* for 30 min at 4 °C, this procedure was repeated again by adding ammonium sulfate (360 mg/mL) to supernatant. The final pellet was resuspended in 20 mM Tris, pH 7.6 and loaded onto a DEAE column, finally eluted in a gradient of 0-1000 mM NaCl. Protein purity was analyzed by SDS-PAGE. For CT-α-syn and CT-A90C, cells were resuspended in 1 mM PMSF, 50 mM MES, pH 6.0 and lysed. Cell lysates were centrifuged at 48,000*g* for 30 min at 4 °C. The supernatant was introduced into a SP cation exchange column (GE Healthcare) and the column eluted in a gradient of 0-1000 mM NaCl. Fractions containing the proteins were concentrated after SDS-PAGE analysis and then chromatographed on a Superdex 75 26/600 column (GE Healthcare) eluted with 50 mM MES and 200 mM NaCl, pH 6.0.

The human PDI gene was cloned into a pET-28a vector containing a His6-tag and a TEV protease cleavage site at the N terminus. The recombinant vector was transformed into *E. coli* BL21 (DE3). The cells were cultured in LB medium at 37 °C. When cell density reached an OD₆₀₀ of 0.8, PDI expression was induced with 1 mM IPTG for 7 h. Cells harvested by centrifugation were resuspended in 20 mM Tris, 500 mM NaCl, pH 8.0 with freshly prepared 10 mg/L protease inhibitors (Sigma-Aldrich). After lysis and centrifugation, the soluble fraction applied to a HisTrap column (GE Healthcare), which was eluted with a linear gradient of 0–500 mM imidazole. Protein was further purified by anion exchange chromatography with a Resource Q column (GE Healthcare) using 20 mM Tris, pH 8.0 and eluted by a linear gradient of 0-1000 mM NaCl, followed by size-exclusion chromatography with a Superdex 200 16/600 column (GE Healthcare) equilibrated with 20 mM Tris, 300 mM NaCl, pH 8.0. Protein concentration was determined either by absorbance of aromatic amino acids at 280 nm with a NanoDrop spectrophotometer (extinction coefficients is 45.38, 5.96 and 1.49 for PDI, FL- and CT-α-syn, respectively), or by Bradford assay using BSA as a standard. Purified proteins were desalted using a HiTrap column (GE Healthcare), lyophilized and stored at −80 °C.

**Liposome preparation**. POPC, POPA, POPS and CL were purchased from Avanti Polar Lipids (Alabaster, AL) and used without purification. Liposomes containing 50% POPC and 50% POPA, and liposomes containing 70% POPC and 30% CL were prepared as described[91,92]. Briefly, lipid powders were mixed and dissolved in CHCl₃. Solution of lipid mixtures were dried, and lipids were hydrated with liposome buffer (20 mM HEPES, 100 mM KCl, pH7.0) to a final concentration of 24 mM. After that, the lipids were sonicated for 10 min with 30% output to prepare liposomes.

**Liposome coflotation assay**. The assay was performed as described[93,94]. Briefly, 20 μM FL- and CT-α-syn were individually incubated with 6 mM liposomes in a total volume of 375 μL liposome buffer (20 mM HEPES, 100 mM KCl, pH7.0) for 30 min at room temperature. After incubation, 375 μL iohexol [80% (w/v)] in

liposomes buffer was added to protein-liposomes mixture (40% final density). In a thick-wall centrifugation tube (Beckman), 300 μL of liposome buffer was on top with 375 μL of 30% iohexol and 375 μL of 35% iohexol underneath, 750 μL of 40% iohexol was placed at the bottom. A SW60Ti rotor (Beckman) was used for gradient centrifugation at 280,000 g for 150 min at 4 ℃ to separate lipid-bound and unbound proteins. After that, liposomes and liposomes-associated proteins floated to the top of the gradient. The mixture was then deconstructed from the top to the bottom into nine equal fractions. Each fraction was mixed with 5 × SDS loading buffer, and 10 μL was subjected to SDS-PAGE. To better resolve FL- or CT-α-syn bands, lipids were allowed to migrate out of the gel as much as possible. Image J software (National Institutes of Health) was used to quantify band intensity.

**¹H-¹⁵N Heteronuclear single quantum coherence**. Spectra from membrane-binding experiments were acquired using 0.1 mM ¹⁵N-enriched FL- and CT-α-syn dissolved in NMR buffer (20 mM HEPES, 100 mM KCl, pH7.0) plus 10% (v/v) D₂O in the absence or presence of 12 mM liposomes. ¹H-¹⁵N HSQC spectra from PDI-binding experiments were acquired using 0.3 mM ¹⁵N-enriched FL- and CT-α-syn dissolved in 20 mM HEPES, 100 mM NaCl, 5 mM tris (2-carboxyethyl) phosphine, pH 7.0, 10% (v/v) D₂O in the absence, or presence of PDI at mole ratios from 10:1 to 1:1. Experiments were carried out at 15 ℃ on a Bruker Avance 800 or 850 MHz NMR spectrometer. Resonance assignments for α-syn are available from the BioMagResBank (entry number 16543). Data were analyzed with Sparky software. Intensity ratios were calculated by peak heights in the presence and absence of liposomes or PDI. CSPs of backbone amides were calculated according to Eq. (1)[95], where $\Delta\delta_H$ and $\Delta\delta_N$ denote the chemical shift difference in the absence and presence of PDI in the ¹H and ¹⁵N dimension, respectively.

$$\Delta\delta = \sqrt{(\Delta\delta_H)^2 + 0.04 \cdot (\Delta\delta_N)^2} \qquad (1)$$

$K_D$ was calculated using the fitting function shown in Eq. (2)[96]:

$$\Delta\delta_{obs} = \frac{\Delta\delta_{max}\left\{([P]_t+[L]_t+K_d)-\left[([P]_t+[L]_t+K_d)^2-4[P]_t[L]_t\right]^{1/2}\right\}}{2[P]_t} \qquad (2)$$

where $\Delta\delta_{obs}$ is the observed FL- or CT-α-syn chemical shift minus the free FL- or CT-α-syn shifts, $[P]_t$ represents the concentration of FL- or CT-α-syn (300 μM), and $[L]_t$ represents PDI concentration, from 0 to 300 μM.

**Paramagnetic relaxation enhancement**. For spin label (MTSL) conjugation, single cysteine variants of ¹⁵N-enriched FL- and CT-α-syn (FL- and CT-A90C) were dissolved in NMR buffer and reduced with 5 mM DTT for 30 min. The proteins were then reacted with a 5-fold mole excess of MTSL at 4 ℃ for 16 h in the dark. Excess MTSL was removed by eluting samples over a HiTrap desalting column (GE Healthcare) into NMR buffer. Complete labeling was confirmed by LC-MS. Spin-labeled FL- and CT-α-syn were lyophilized or used directly. Intra-molecular PRE experiments required 100 μM ¹⁵N-MTSL-labeled protein. PRE experiments were conducted in NMR buffer containing 10% (v/v) D₂O at 15 ℃ using a Bruker Avance 850-MHz spectrometer. Diamagnetic samples were prepared by adding a 10-fold mole excess of ascorbic acid to the paramagnetic samples. PRE data were processed using NMRpipe-software and analyzed with Sparky-software. PRE effects were quantified as the ratios of peak intensities recorded in the paramagnetic state versus the diamagnetic state, respectively. The transverse relaxation rate enhancements ($R_2$) could be calculated with Eq. (3)[97]:

$$R_2 = \frac{1}{T_{b-}T_a} \ln \frac{I_{dia}(T_b)I_{para}(T_a)}{I_{dia}(T_a)I_{para}(T_b)} \qquad (3)$$

where $I_{dia}$, $I_{para}$ denote NMR signal intensity in the diamagnetic and paramagnetic state, respectively, the interval time $T_b-T_a$, was 17.2 ms for FL-α-syn and 20 ms for CT-α-syn.

**Mitochondrial membrane potential**. SK-N-SH cells (purchased from Procell, CL-0214) were incubated with 50 μM FL- and CT-α-syn in growth medium at 37 ℃ for 5 h. The JC-1 kit (Sigma-Aldrich) provided the prepared reagents and was used according to manual. Images were acquired with a confocal fluorescence microscope (Leica TCS SP8). Fluorescence intensity was analyzed with Image J software.

**Cytochrome c release**. SK-N-SH cells were cultured in MEM supplemented with 10% fetal bovine serum and antibiotics in a humidified atmosphere of 5% CO₂ at 37 ℃. At least 10⁷ SK-N-SH cells were used per experiment. The mitochondria were prepared with a Mitochondria Isolation Kit (Solarbio) for cultured cells. Freshly isolated mitochondria were resuspended in ice cold buffer (10 mM HEPES, 2 mM K₂HPO₄, 10 mM succinate, 250 mM sucrose, 1 mM ATP, 0.08 mM ADP, 1 mM DTT, pH 7.5)[56]. Aliquots were incubated with FL- or CT-α-syn at 37 ℃ for 1 h. After incubation, the mixture was centrifuged at 13000 g at 4 ℃ for 10 min. An equal volume of the supernatant was subjected to SDS-PAGE and the band transferred to a nitrocellulose membrane. The membrane was blocked in TBS-T (20 mM Tris, 150 mM NaCl, 0.05% Tween-20, pH 8.0) with 5% skim milk overnight at 4 ℃ prior to the adding anti-cyto c monoclonal antibody (Abcam, Cat.ab133504, 1:1000 dilution) and incubating for 1 h at room temperature.

Following three washes with TBS-T, the membrane was incubated with goat anti-rabbit-HRP secondary antibody in blocking buffer (Beijing Biodragon, Cat.BF03008, 1:5000 dilution) in TBS-T with 5% skim milk for 1 h at room temperature. The membrane was visualized by enhanced chemiluminescence reagents (BioRad) after another three washes.

**Dynamic light scattering**. FL- or CT-α-syn (20 μM) were incubated with 100 μM liposomes in a total volume of 1 ml of NMR buffer at room temperature for 30 min. Clustering activity was measured by dynamic light scattering using a Protein Solutions DynaPro instrument (Malvern) in triplicate, with an average of 10 data points. Particle size distribution of liposomes for control experiments was monitored under same conditions.

**Fibrillation monitored by thioflavin T fluorescence**. FL- or CT-α-syn (0.3 mM) was mixed with or without 0.3 mM PDI in 20 mM Tris, 100 mM NaCl, 2 mM tris (2-carboxyethyl) phosphine, pH 7.5, 0.01% (w/v) NaN₃ and filtered through a 0.22 μm Millex filter. Samples (200 μL) containing 15 μM ThT were dispensed along with a 2.5-mm glass bead in each well of a black, transparent bottomed, 96-well plate. The plate was sealed with a Al sealing tape (Corning) and incubated in a SpectroMax i3x microplate reader (Molecular Devices) at 37 ℃ with 500 rpm high-grade orbital shaking. ThT fluorescence was measured every 10 min using excitation and emission wavelengths of 444 nm and 482 nm, respectively. The data were analyzed using Eq. (4)[10]:

$$F(t) = F_0 + A/(1 + e^{-k(t-t_{1/2})}) \qquad (4)$$

$F(t)$ was normalized by dividing the largest value in each experiment, $k$ represents the fibrillation rate constant and $t_{1/2}$ is the aggregation half-time. The lag time $t_{lag}$ was calculated as $t_{lag} = (t_{1/2} - 2/k)$.

**Statistics and reproducibility**. Data were expressed as mean ± SEM from three to four independent experiments. Bar graphs were shown in dot-plot format showing data distribution. Statistics were analyzed by one-way ANOVA and $*p < 0.05$, $**p < 0.01$ and $***p < 0.001$ were considered as significant differences. The number of replicates were described in the figure legends.

**Reporting summary**. Further information on research design is available in the Nature Research Reporting Summary linked to this article.

## Data availability
The full and uncropped gel/blot images are shown in Supplementary Fig. 4. All data generated or analyzed during this study are included in the article and Supplementary Data 1–3.

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

## Acknowledgements

This work is supported by the National Key R&D Program of China (2018YFE0202300), the National Natural Sciences Foundation of China (21673284, 21925406 and 21991080) and K. C. Wong Education Foundation.

## Author contributions

C.Z., Y.P., Z.Z. L.J., G.J.P., X.Z., M.L., and C.L. conceived the study. C.Z., Y.P., Z.Z., L.X., and X.L. performed the experiments. C.Z., Y.P., Z.Z., L.J., G.J.P., X.Z., M.L., and C.L. analyzed the data. C.Z., Z.Z., L.J., G.J.P., X.Z., M.L., and C.L. wrote the manuscript.

## Competing interests

The authors declare no competing interests.
