## [Peer Review File · Communications Biology]

Reviewers' comments:

Reviewer #1 (Remarks to the Author):

In the manuscript titled "α-synuclein C-terminal truncation modulates its cytotoxicity and aggregation by promoting its N-terminal interactions with membrane and chaperone" Cai Zhang et al. have elaborated the modulatory role of the C-terminus of the amyloidogenic α-synuclein protein. The study involving solution state NMR spectroscopy along with few biophysical techniques discusses, in a very easy to comprehend language, the pathological role of C-terminal truncation in PD pathogenesis- particularly concerning the membrane association of the protein and the chaperone interaction.

A few minor concerns that might help in the improvement of the content are detailed below:

1. A few sentences would benefit upon addition of the corresponding references. A few examples are:

- The section on "Significance Statement" contains the statement "15% of which is C-terminal truncated (CT-α-syn) in pathological inclusions." -requires proper referencing.
- Similarly, on page #4, line 91/92, the statement "In cells, the N-terminal domain interacts with membranes and chaperones that modulates fibrillation" requires referencing.

2. Line 102, the authors state "The interaction may arise from the more extend conformation of CT-α-syn". This has been described by previous reports that state the "open" conformation of the protein as opposed to the closed conformer where the two termini are in long-range interactions, for example, Chem. Commun., 2018,54, 3605-3608

3. The authors discuss the membrane associated detrimental functioning of the CT-α-syn in association with the mitochondrial membrane. It might be interesting to see the effect of the truncation on the functional association of the protein with the synaptic vesicular membranes.

4. Rapid membrane-associated fibrillation can corroborate with a reduced cytotoxicity to the neuronal cells- as the oligomeric intermediates are often elaborated to be the most toxic forms. Can the authors comment on the cytotoxicity at a reasonable concentration in cells?

5. Previous studies have discussed the role of salts in the buffer that induces a direct effect in the proteins conformation and the opening-and closing in direct association with the aggregation propensity and the membrane association (Protein Sci. 2008 Aug; 17(8): 1434-1445). Since the authors have used 100mM NaCl in their buffer solutions, does that affect the overall change in the FL-α-syn functional attribute?

6. Line 234, the authors claim "diminishes the inhibitory effect of PDI on CT-α-syn aggregation" however, with the provided data, this statement cannot be justified. Hence can only be stated as the hypothesis in Line 236. Similarly, in the sentence starting in line 253, the authors claim "...cyto c release and buildup of mitochondrial reactive oxygen species (mtROS), activating the apoptosome and NLRP3 inflammasome." However, the data discussed herein is not sufficient to justify this statement. While, it is plausible to conclude that the strong association of the CT- variant of the protein to the mitochondrial membranes can cause this cascade of events (based on the previous studies as referred in the context), the current study does not provide the experimental evidence to conclusively discuss this sequence of events. Therefore, it is essential that the authors consider re-writing and rewording of the section.

7. Line 308, the method for preparation of the liposomes, the authors state "were prepared as described (79, 80).", it is recommended that the authors briefly discuss the methodology in a line or two for better readership.

8. To this reviewer, the figure 5, elaborates the mechanism of CT- α-syn in the PD pathogenesis, including the functional association of the protein for the synaptic vesicles. However, the studies

discussed herein, do not elaborate the mechanism of pathogenesis and it should be better if the authors reconsider the figure such that it best summarizes the study and its findings- that although not conclusive of the entire mechanism, but has enough merit to itself.

Reviewer #2 (Remarks to the Author):

In this manuscript, Zhang et al. probe the impact of C-terminal residues in full-length alpha synuclein (FL- α -syn) in membrane/chaperone interaction using solution-NMR and cytotoxicity assessed by loss of membrane potential in mitochondria and cytochrome c release. Using a C-terminal truncated variant of α -syn (CT- α -syn), the authors show that C-terminal truncation results in a more extended and exposed conformation of α -syn using solution-NMR, and claim that is responsible for the enhanced affinity to model membranes and a chaperone PDI. They further carry out experiments to show that C-terminal truncation results in faster aggregation compared to FL- α -syn and causes mitochondrial damage; which is already well-known (Sorrentino et al. 2018, Ma et al. 2018, Krzystek et al 2021). The conclusions in the manuscript are partly original. The novel aspect is that C-terminal truncation results in a more extended conformation of α -syn and the differential inhibitory effect of PDI on FL- α -syn and CT- α -syn fibrillation. The work is not entirely convincing in my opinion since some of the conclusions are not supported thoroughly by experimental evidence. I provide suggestions for additional experiments/clarifications to the authors hereunder to strengthen their conclusions.

1. The authors show in Figure 1C-D, a liposome floatation assay with POPC:POPA (1:1) and that \sim 80% of CT- α -syn binds the membrane whereas \sim 50% of FL- α -syn is bound state (Figure 1C-D), indicating a stronger association of CT- α -syn to POPC: POPA (1:1) liposomes. But strangely, the HSQC measurements are done with POPC:CL (9:1 and 7:3). I do not understand this. The authors need to show HSQC measurements with POPC: POPA (1:1) liposomes or show liposome floatation results with POPC: CL (9:1) in Figure 1. I do see two SDS-PAGE images in Figure S1, but the legend is unclear to discern if they represent POPC:CL (9:1), or POPC: POPA (1:1) fractions. The SDS-PAGE data accompanying the floatation assay lacks information. How much protein was loaded in each case? the CT- α -syn has a weaker band intensity in general, which may arise due to its small size if equimolar concentrations of proteins were used. Are the images contrasted equally?

2. The authors need to describe the liposome co-floatation assay in detail as I find crucial information missing. There is some mismatch in Figure 1B and the methods describing the liposome co-floatation assay. The authors mention, "The sample in 40% (w/v) iohexol was transferred to a thick-wall centrifugation tube (Beckman) before 300 μ l of liposome buffer, 375 μ l of 30% (w/v) iohexol and 375 μ l of 35% (w/v) iohexol were added in that order without mixing". Shouldn't 35% (w/v) iohexol be added before 30% (w/v) iohexol as per Figure 1B? Additionally, the authors mention lipid-bound proteins were recovered from 0% and 40% iohexol. However, there should be no lipid-bound proteins at 40% iohexol, and only unbound proteins must be present here, according to their figure. What was the protein: lipid concentration used for floatation assay? What was a buffer composition, temperature, and pH used? The temperature is extremely crucial since α S binding to liposomes is strongly affected by T_m , and the use of cardiolipin may cause the T_m to change depending on the temperature used. These factors are relevant to α S binding to liposomes.

3. From Figure 1E and Figure S1B, the authors show that signals for CT- α -syn are more attenuated in the presence of liposomes containing cardiolipin (CL), indicating a stronger binding of CT- α -syn to acidic lipids than FL- α -syn. I am not convinced entirely. First, I would like to know the justification for using 50% POPA, a lipid not found beyond 3% in synaptic vesicles (Takamori et al. 2016, Cell.; Deutsch, J.W. et al, 1981. Biochem; Binotti et al. 2021, Arch.Biochem. Biophys.). Why was POPS, a similar anionic lipid but with a larger headgroup than the cone-shaped POPA, not used? POPA lipids, due to their small headgroup, trigger a negative membrane curvature leading to packing defects, especially when SUVs are used, as in this study. Packing defects are also promoted by lipids with unsaturated acyl chains and/or small head groups like POPA (Bigay et al. 2012, Dev. Cell.) and

cardiolipin (CL). The presence of CL can provide a driving force towards an increase in membrane negative curvature stress (Lewis et al. 2009, BBA Biomembr.). Packing defects and curvature stress expose hydrophobic regions in liposomes will obviously "attract" the extended CT- α -syn with a more exposed N-terminus to liposomes. Using SUVs with lipids prone to induce curvature and packing defects makes it tough to judge if truncating the C-terminal region facilitates membrane binding or the exposed hydrophobic regions in the liposome facilitate membrane binding. The authors suggest the driving force is the removal of the C-terminal region but packing defects in the used liposomes is equally a plausible reason for their observations. If removal of the C-terminal region increases the affinity of CT- α -syn for lipid membranes, measurements with POPS liposomes (preferably LUVs that have reduced curvature stress) should show similar results. To strengthen their hypothesis, the authors must demonstrate measurements with POPS or a non-curvature promoting lipid.

4. Page 8: In the JC-1 staining and cyto c release experiments, the author claim that show CT- α -syn collapses the potential, which coincides with cyto c release (Figure 4C). By contrast, FL- α -syn does not impact cyto c release or membrane potential. While the observation is compelling, it could also be explained by a faster aggregation of CT- α -syn (See Fig 4A, E) compared to FL- α -syn without invoking a stronger membrane binding argument. A very recent paper by Schierle group at Cambridge shows precisely that the more exposed the N-terminus and the beginning of the NAC region of α Syn are, the more aggregation prone monomeric α Syn conformations become. Further, the aggregation process of FL- α -syn is enough to damage membranes is well known (Chaudhary et al. 2014 (FEBS Lett)). The JC-1 staining in Fig 4A is after 5 h treatment with 50 μ M FL- α -syn and CT- α -syn at 37 $^{\circ}$ C and judging from Fig 4E, aggregation of CT- α -syn is well beyond lag phase. So how can the authors be sure the proteins are still monomeric? It is also unclear what protein concentrations were used in Fig 4E, making the comparison difficult. The buffer conditions for JC-1 staining are also missing. Can the authors show that there are no early order aggregates present that may damage SK-N-SH cells or cause cyto c leakage? I would like to see a size-exclusion profile of 50 μ M FL- α -syn and CT- α -syn after 5 h incubation with SK-N-SH cells to accept this hypothesis. It seems equally plausible that the high concentration of FL- and CT- α -syn used may have aggregated already into oligomeric intermediates, collapsed the potential, and caused cyto c release.

5. Minor points:

- a. The authors should try to explain the methods (in figure legends or method section) more in detail. The approaches used in the paper are pretty nice but often leave room for multiple interpretations due to lack of information.
- b. I cannot find the ThT concentration used in aggregation experiments. ThT itself may trigger aggregation of FL- α -syn if used at high concentrations. The authors mention the use of 1.5 mM stock, but actual concentration details are missing.
- c. On page 10, in the liposome preparation section, the authors mention C1 and C3 liposomes. There is no mention of these in the manuscript, and I am unsure what they refer to.
- d. "mole ration" is probably mole ratio on page 10.

Reviewer #3 (Remarks to the Author):

Li and coworkers compare full-length and C-terminally truncated α -synuclein (α -syn) in terms of interactions with anionic membranes and the chaperone protein disulfide isomerase (PDI), the compactness of the protein (by NMR) and impact on cell biology. They highlight some interesting observations and by and large their conclusions seem reasonable and build up a compelling story. However, I have a number of reservations about the current version of the manuscript. In a number of cases, I am unable to follow the analysis of the authors:

1. Fig. 2D: I am unable to see any significant change in the binding isotherms of FL and CT α -syn based on these graphs – certainly not an order of magnitude difference. A K_D of ca 40 μ M should lead to saturation by 0.3 mM and that is not the case at all. Also, the authors should label the curves with the associated residues (as in Fig. S4C) and provide the K_D values of the individual fits to see the

spread.

2. The same issue about KD values is relevant for Fig. S4C where I cannot see any significant difference. Also, are the reported KD values an average based on the individual curve fits?
3. Fig. 3: I find it difficult to see major differences between panels A and B (also confounded by the difference in length of the two constructs). I would not conclude anything about significant changes from the presented data. Also, why is Tyr39 singled out? Is that the supposed contact point for Cys90 MTSL? (It is not particularly broadened).
4. It is very odd that the authors introduce new results in the Discussion (Fig. S4A-C). This should be included in the appropriate place in Results and be integrated into the narrative there.
5. PDI seems an appropriate chaperone to investigate despite the lack of Cys residues (and disulfide bonds) in α -syn; however, additional references could be provided eg. PMID 33516852 and 32850841. However, it is not clear if the conclusions are general. The study would benefit greatly from inclusion of at least one other chaperone, e.g. Hsp70, which is considered a central chaperone in the whole PD field. Simply collecting data corresponding to Fig. 2 (at the very least at one Hsp70 concentration, but preferably more to gauge changes in affinity) would be a major improvement.

Minor issues:

1. The authors should state clearly in Introduction/Results that CT α -syn contains residues 1-99. This information is currently buried in the M&M section.
2. Minor linguistic issues: "interfering the interaction", "inhibition on fibrillation", "C-terminal truncated", "50% of FL- α -syn is bound state"
3. There is nothing wrong with the sentence "These observations raise the question of whether C-terminal truncations affect the interaction of the N-terminal domain with membranes and chaperones, which could affect its functionality" but given the conclusions from the manuscript and the preceding statement about C- and N-terminal interactions, I would suggest to rephrase to e.g. "These observations raise the question of whether C-terminal truncations free up the N-terminal domain to interact [more extensively?] with membranes and chaperones, thus affecting α -syn functionality".
4. Fig. S1A only differs from Fig. 1 in having a slightly different lipid composition (30% CL rather than 10%). This should be emphasized more clearly in the beginning of the figure legend to both Fig. 1 and Fig. S1. In that context "Data for CT- α -syn show that signals are more attenuated in the presence of liposomes containing CL" should be clarified to indicate that it is 30 versus 10% (there are no data for 0% CL).
5. Fig 4A: why is there such a significant green signal for control as well? Is apoptosis expected to be wide-spread even in normal cells or are the cells stressed because of growth conditions? (I ask out of ignorance of practical cell biology).
6. Fig 4C: Can these data be densitometrically quantified? Difficult to assess as is.
7. Fig. 4D: The blank controls are almost impossible to see (which of course emphasizes its similarity with FL- α -syn but should still be redrawn).
8. There are no references to Figs. S2 and S3 in the text. The conclusion in Fig. S2 about a more extended conformation for CT α -syn needs to be elaborated (it is difficult for the untrained eye to see if the difference to FL α -syn is significant).

Dear Prof. Chattopadhyay,

Thank you for processing “ α -Synuclein C-terminal truncation modulates its cytotoxicity and aggregation by promoting its N-terminal interactions with membrane and chaperone”, which is exclusively submitted to *Communications Biology*. We have performed several additional experiments and revised the manuscript according to the Reviewers’ comments. The corrections are marked red. Our point-by-point response follows.

Response to Reviewer #1

Q1. A few sentences would benefit upon addition of the corresponding references. A few examples are:

- The section on “Significance Statement” contains the statement “15% of which is C-terminal truncated (CT- α -syn) in pathological inclusions.” -requires proper referencing.
- Similarly, on page #4, line 91/92, the statement “In cells, the N-terminal domain interacts with membranes and chaperones that modulates fibrillation” requires referencing.

A1: Done. Ref 1, 37 and 38 were added.

Q2. Line 102, the authors state “The interaction may arise from the more extend conformation of CT- α -syn”. This has been described by previous reports that state the “open” conformation of the protein as opposed to the closed conformer where the two termini are in long-range interactions, for example, *Chem. Commun.*, 2018,54, 3605-3608.

A2: We changed “extended” to “open” throughout, and added reference 45.

Q3. The authors discuss the membrane associated detrimental functioning of the CT- α -syn in association with the mitochondrial membrane. It might be interesting to see the effect of the truncation on the functional association of the protein with the synaptic vesicular membranes.

A3: We also observe enhanced association of CT- α -syn with synaptic vesicular-mimicking membranes composed of POPC/cholesterol/DOPS (45:40:15) by NMR spectra, which is evident by reduction CT- α -syn signal compared to that of FL- α -syn (Figure R1 below). Its functional consequences would be another interesting topic.

Figure R1. Overlaid ^1H - ^{15}N HSQC spectra of 0.125 mM ^{15}N -enriched FL- (left) and CT- α -syn (right) in the absence (black) and presence (red) of 12 mM synaptic vesicular mimicking membranes composed of POPC/cholesterol/DOPS (45:40:15). Residue-resolved NMR signal attenuation (I/I_0) of FL- (black) and CT- α -syn (red) upon adding membranes are shown below the corresponding ^1H - ^{15}N HSQC spectra. Values <1.0 indicate interaction.

Q4. Rapid membrane-associated fibrillation can corroborate with a reduced cytotoxicity to the neuronal cells- as the oligomeric intermediates are often elaborated to be the most toxic forms. Can the authors comment on the cytotoxicity at a reasonable concentration in cells?

A4: We did not detect fibrils during 5 h incubation with unshaken cells. First, ThT fluorescence shows no changes with 50 μM and 300 μM FL-and CT- α -syn at 37 $^\circ\text{C}$ in buffer (20 mM HEPES, 100 mM KCl, pH7.0) (Figure R2A), indicating that 50 μM FL- and CT- α -syn are unable to form fibrils under static conditions within 5-10 h. Second, we used detergent-solubility testing to probe whether 50 μM FL- or CT- α -syn forms fibrils in the presence of cells in 5 h (Klucken J, et al. 2004, J. Biol. Chem. 279(24): 25497-502;

Dutta D, et al. 2021, Nat. Commun. 12(1): 5382). The immunoblot assay shows that α -syn only can be detected in the Triton X-100 soluble fraction, suggesting no fibril formation (Figure R2B). Meanwhile, we do not detect endogenous α -syn in control samples probably because of the low cell numbers ($\sim 1.5 \times 10^4$) in the 6-well plate. Under our conditions, CT- α -syn exerts cytotoxicity even at concentration less than 50 μ M, mainly through enhanced monomer-membrane interactions.

Figure R2. (A) Aggregation of FL- and CT- α -syn with and without agitation within 10 h by ThT fluorescence assay at 37 °C. **(B)** Immunoblots of detergent-soluble and insoluble α -syn after 5 h incubation with cells. (Samples prepared as described by Klucken J, et al. 2004, J. Biol. Chem. 279(24): 25497-502; Dutta D, et al. 2021, Nat. Commun. 12(1): 5382). After 5 h incubation, Triton X-100 is added to FL- or CT- α -syn to a final concentration of 1% and incubated on ice for 30 min followed by centrifugation at 15000g for 15 min at 4 °C. The supernatant is referred to as the soluble fraction. The pellet is then resuspended in loading buffer (2% SDS), washed with PBS, and boiled for 15 min (insoluble fraction).)

Q5. Previous studies have discussed the role of salts in the buffer that induces a direct effect in the proteins conformation and the opening-and closing in direct association with the aggregation propensity and the membrane association (Protein Sci. 2008 Aug; 17(8): 1434–1445). Since the authors have used 100mM NaCl in their buffer solutions, does that affect the overall change in the FL- α -syn functional attribute?

A5: We agree that salts impact the conformational ensemble of α -syn. Our study focuses on the conformational and functional differences between FL- and CT- α -syn under the same condition. The results in vitro and in cells show that the C-terminal truncation promotes N-terminal interactions with membrane and chaperone.

Q6. Line 234, the authors claim “diminishes the inhibitory effect of PDI on CT-a-syn aggregation” however, with the provided data, this statement cannot be justified. Hence can only be stated as the hypothesis in Line 236. Similarly, in the sentence starting in line 253, the authors claim “...cyto c release and buildup of mitochondrial reactive oxygen species

(mtROS), activating the apoptosome and NLRP3 inflammasome.” However, the data discussed herein is not sufficient to justify this statement. While, it is plausible to conclude that the strong association of the CT- variant of the protein to the mitochondrial membranes can cause this cascade of events (based on the previous studies as referred in the context), the current study does not provide the experimental evidence to conclusively discuss this sequence of events. Therefore, it is essential that the authors consider re-writing and rewording of the section.

A6: We have reworded the section. “We hypothesize that the stronger interaction of PDI with segment 1 of CT- α -syn explains the diminished inhibitory effect on aggregation.” Lines 213-214.

“Moreover, the stronger interaction between the CT- α -syn and mitochondrial membrane collapses the mitochondrial membrane potential and enhances membrane permeability, causing cyto c release. Based on previous studies and our results, we proposed that the mitochondrial dysfunction likely activates the apoptosome and the NLRP3 inflammasome (80-85), which may lead to more truncation of FL- α -syn, forming a negative-feedback loop that amplifies truncation and elicits cell death (86-88).” Lines 278-284.

Q7. Line 308, the method for preparation of the liposomes, the authors state “were prepared as described (79, 80).”, it is recommended that the authors briefly discuss the methodology in a line or two for better readership.

A7: We have added detail about liposomes preparation: “Briefly, lipid powders were mixed and dissolved in CHCl_3 . Solution of lipid mixtures were dried, and lipids were hydrated with liposome buffer (20 mM HEPES, 100 mM KCl, pH7.0) to a final concentration of 24 mM. After that, the lipids were sonicated for 10 min with 30% output to prepare liposomes.” Lines 330-333.

Q8. To this reviewer, the figure 5, elaborates the mechanism of CT- α -syn in the PD pathogenesis, including the functional association of the protein for the synaptic vesicles. However, the studies discussed herein, do not elaborate the mechanism of pathogenesis and it should be better if the authors reconsider the figure such that it best summarizes the study and its findings- that although not conclusive of the entire mechanism, but has enough merit to itself.

A8: We have replotted the data in Figure 5.

Response to Reviewer #2

Q1. The authors show in Figure 1C-D, a liposome floatation assay with POPC:POPA (1:1) and that ~ 80% of CT- α -syn binds the membrane whereas ~50% of FL- α -syn is bound state

(Figure 1C-D), indicating a stronger association of CT- α -syn to POPC: POPA (1:1) liposomes. But strangely, the HSQC measurements are done with POPC:CL (9:1 and 7:3). I do not understand this. The authors need to show HSQC measurements with POPC: POPA (1:1) liposomes or show liposome floatation results with POPC: CL (9:1) in Figure 1. I do see two SDS-PAGE images in Figure S1, but the legend is unclear to discern if they represent POPC:CL (9:1), or POPC: POPA (1:1) fractions. The SDS-PAGE data accompanying the floatation assay lacks information. How much protein was loaded in each case? The CT- α -syn has a weaker band intensity in general, which may arise due to its small size if equimolar concentrations of proteins were used. Are the images contrasted equally?

A1: We have clarified this issue. Results of the co-flotation assay and HSQC measurements with POPC/CL (7:3) liposomes are now shown in Figure 1. We moved the POPC/POPA (1:1) liposome data to Figure S1. The POPC/CL (9:1) liposome data have been removed because POPC/CL (7:3) liposomes also resemble mitochondrial membranes. The two SDS-PAGE images in Figure S1 are additional co-flotation results of POPC/POPA (1:1).

We use the same concentration (50 μ M) of FL- and CT- α -syn for the co-flotation assay. The gel is stained with Coomassie Brilliant Blue R250. The weaker band intensity of CT- α -syn may be due to its lack of the C-terminus, which has most of the aromatic residues. The contrasts is uniform across SDS-PAGE images.

Q2. The authors need to describe the liposome co-flotation assay in detail as I find crucial information missing. There is some mismatch in Figure 1B and the methods describing the liposome co-flotation assay. The authors mention, "The sample in 40% (w/v) iohexol was transferred to a thick-wall centrifugation tube (Beckman) before 300 μ l of liposome buffer, 375 μ l of 30% (w/v) iohexol and 375 μ l of 35% (w/v) iohexol were added in that order without mixing". Shouldn't 35% (w/v) iohexol be added before 30% (w/v) iohexol as per Figure 1B? Additionally, the authors mention lipid-bound proteins were recovered from 0% and 40% iohexol. However, there should be no lipid-bound proteins at 40% iohexol, and only unbound proteins must be present here, according to their figure. What was the protein: lipid concentration used for flotation assay? What was a buffer composition, temperature, and pH used? The temperature is extremely crucial since aS binding to liposomes is strongly affected by T_m , and the use of cardiolipin may cause the T_m to change depending on the temperature used. These factors are relevant to aS binding to liposomes.

A2: We have added details to Materials and Methods (Lines 334-344).

To load the gradient, we utilize a specific injector to place the denser iohexol at the bottom of the centrifuge tube, which avoids disrupting the interface between the high-density and low-density iohexol and forms a density gradient (Grieger JC et al. 2006, Nat Protoc. 1(3): 1412-28).

We agree α -syn -liposome binding is strongly affected by T_m. We use three phospholipids: POPC, POPA and POPS, with T_m values of -2°C, 28°C and 12°C, respectively (Jiang Z et al. 2015, J. Phys. Chem. B. 119(14): 4812-23). The incubation temperature is ~25°C. Cardiolipin can change T_m, a key factor that affects α -syn binding to liposomes. For POPC/CL (7/3) mixture, adding CL increases the T_m of POPC (-2°C) to 16 °C (S.C. Lopes et al. 2009, *BBA Biomembr.* 1860(11): 2465-2477). Although T_m and membrane composition affects α -syn binding, it does not affect our conclusion because we focused on the conformational and functional differences between FL- and CT- α -syn. The degree of the difference may depend on lipid composition, pH, temperature, buffer, etc.

*Q3. From Figure 1E and Figure S1B, the authors show that signals for CT- α -syn are more attenuated in the presence of liposomes containing cardiolipin (CL), indicating a stronger binding of CT- α -syn to acidic lipids than FL- α -syn. I am not convinced entirely. First, I would like to know the justification for using 50% POPA, a lipid not found beyond 3% in synaptic vesicles (Takamori et al. 2016, *Cell.*; Deutsch, J.W. et al, 1981. *Biochem*; Binotti et al. 2021, *Arch.Biochem. Biophys.*). Why was POPS, a similar anionic lipid but with a larger headgroup than the cone-shaped POPA, not used? POPA lipids, due to their small headgroup, trigger a negative membrane curvature leading to packing defects, especially when SUVs are used, as in this study. Packing defects are also promoted by lipids with unsaturated acyl chains and/or small head groups like POPA (Bigay et al. 2012, *Dev. Cell.*) and cardiolipin (CL). The presence of CL can provide a driving force towards an increase in membrane negative curvature stress (Lewis et al. 2009, *BBA Biomembr.*). Packing defects and curvature stress expose hydrophobic regions in liposomes will obviously "attract" the extended CT- α -syn with a more exposed N-terminus to liposomes. Using SUVs with lipids prone to induce curvature and packing defects makes it tough to judge if truncating the C-terminal region facilitates membrane binding or the exposed hydrophobic regions in the liposome facilitate membrane binding. The authors suggest the driving force is the removal of the C-terminal region but packing defects in the used liposomes is equally a plausible reason for their observations. If removal of the C-terminal region increases the affinity of CT- α -syn for lipid membranes, measurements with POPS liposomes (preferably LUVs that have reduced curvature stress) should show similar results. To strengthen their hypothesis, the authors must demonstrate measurements with POPS or a non-curvature promoting lipid.*

A3: This comment is similar to #2. We focus on the difference between FL- and CT- α -syn under the identical conditions. We agree that α -syn binds membranes with packing defects. Indeed, our NMR data show that α -syn exhibits different affinities for various membranes via its N-terminal domain (Figure R3), while the strong interactions between α -syn and CL or POPA are attributable to both negative charge and packing

defects. As requested, we have performed additional NMR experiments and flotation assays using POPC/POPS (7:3) liposomes (Figure S1A-B and D). According to flotation assay, the membrane-bound proportion is ~50% and ~25% for CT- α -syn and FL- α -syn, respectively. Moreover, NMR signals from CT- α -syn are more attenuated in liposomes containing 30% POPS compared to FL- α -syn, indicating a stronger interaction of CT- α -syn with POPC/POPS (7:3) liposomes as well.

Figure R3. Residue-resolved NMR signal attenuation (I/I_0) of FL- α -syn in liposomes composed of POPC/CL (7:3) (black), POPC/POPA (1:1) (red), POPC/POPS (7:3) (blue) and POPC/cholesterol/DOPS (45:40:15) (purple).

Q4. Page 8: In the JC-1 staining and cyto c release experiments, the author claim that show CT- α -syn collapses the potential, which coincides with cyto c release (Figure 4C). By contrast, FL- α -syn does not impact cyto c release or membrane potential. While the observation is compelling, it could also be explained by a faster aggregation of CT- α -syn (See Fig 4A, E) compared to FL- α -syn without invoking a stronger membrane binding argument. A very recent paper by Schierle group at Cambridge shows precisely that the more exposed the N-terminus and the beginning of the NAC region of aSyn are, the more aggregation prone monomeric aSyn conformations become. Further, the aggregation process of FL- α -syn is enough to damage membranes is well known (Chaudhary et al. 2014 (FEBS Lett)). The JC-1 staining in Fig 4A is after 5 h treatment with 50 μ M FL- α -syn and CT- α -syn at 37 $^{\circ}$ C and judging from Fig 4E, aggregation of CT- α -syn is well beyond lag phase. So how can the authors be sure the proteins are still monomeric? It is also unclear what protein concentrations were used in Fig 4E, making the comparison difficult. The buffer conditions for JC-1 staining are also missing. Can the authors show that there are no early order aggregates present that may damage SK-N-SH cells or cause cyto c leakage? I would like to see a size-exclusion profile of 50 μ M FL- α -syn and CT- α -syn after 5 h incubation with SK-N-SH cells to accept this hypothesis. It seems equally plausible that the high concentration of FL- and CT- α -syn used may have aggregated already into oligomeric intermediates, collapsed the potential, and caused cyto c release.

A4: The experimental conditions for fibrillation and cyto c release cannot be compared directly. The data in Fig 4E were acquired in 300 μ M protein with constant agitation

with glass beads, which accelerates the aggregation, but the data in Fig 4A were acquired under static conditions. As addressed above (Figure R2, referee 1, Q4), we have demonstrated the absence of insoluble inclusions after 5 h incubation with SK-N-SH cells. As the Referee requested, we also used analytical size-exclusion chromatography (SEC) to resolve early oligomers in 50 μ M FL- and CT- α -syn. As shown in Figure R4, FL- α -syn is monomeric after 5 h incubation with cells but CT- α -syn appears in two peaks. Next, we used an immune-dot blot to identify these peaks with anti- α -syn antibody. The former peak, which is possibly an oligomer, shows negative signals, which may arise from the leaked protein due to the toxicity of CT- α -syn. The latter peak is positive for α -syn. The SEC results indicate that CT- α -syn does not form oligomeric intermediates. These results suggest that both FL- and CT- α -syn exist as monomers during the 5 h incubation. Therefore, the cytotoxicity we observed from CT- α -syn at < 5 h is mainly caused by its stronger membrane interaction instead of its toxic aggregates.

The JC-1 dye and buffers for JC-1 staining were from the JC-1 kit.

Figure R4. Size-exclusion chromatography of FL- (black) and CT- α -syn (red) without (dashed line, left) and with incubation (solid line, right). Control (blue solid line) represents incubation without protein.

Q5. Minor points:

Q5a. *The authors should try to explain the methods (in figure legends or method section) more in detail. The approaches used in the paper are pretty nice but often leave room for multiple interpretations due to lack of information.*

A5a: We have clarified the Method section and Figure legends.

Q5b. *I cannot find the ThT concentration used in aggregation experiments. ThT itself may trigger aggregation of FL- α -syn if used at high concentrations. The authors mention the use of 1.5 mM stock, but actual concentration details are missing.*

A5b: The concentration is 15 μ M. We have added the description to the Materials and Methods section.

Q5c. On page 10, in the liposome preparation section, the authors mention C1 and C3 liposomes. There is no mention of these in the manuscript, and I am unsure what they refer to.

A5c: C1 and C3 refer to POPC/CL (9:1) and POPC/CL (7:3) liposomes, respectively. We now provided the lipid composition.

Q5d. "mole ration" is probably mole ratio on page 10.

A5d: Done.

Response to Reviewer #3

Q1. Fig. 2D: I am unable to see any significant change in the binding isotherms of FL and CT α -syn based on these graphs – certainly not an order of magnitude difference. A K_D of ca 40 μ M should lead to saturation by 0.3 mM and that is not the case at all. Also, the authors should label the curves with the associated residues (as in Fig. S4C) and provide the K_D values of the individual fits to see the spread.

A1: We have modified the figures. The K_D values are from fitting the average chemical shift perturbations from all distinguishable residues in segments 1 (V3, M5, G7, L8, S9, K12, V15, A17, A18, A19 and E20) and 2 (T33, V37, L38, V40, G41 and T44) of FL-, CT- and Y39E- α -syn. The concentration of α -syn in the titration is 0.3 mM. Furthermore, the disappearance signals at PDI: α -syn at mole ratios >1:1, prevented acquisition of more data. To make the graph clearer, we added the theoretical curve to a molar ratio of 2:1 (Fig.2D).

Q2. The same issue about K_D values is relevant for Fig. S4C where I cannot see any significant difference. Also, are the reported K_D values an average based on the individual curve fits?

A2: As addressed above, the K_D values are calculated from curve fitting using average CSPs values. The K_D value for segment 1 of Y39E- α -syn:PDI is $50 \pm 20 \mu$ M.

Q3. Fig. 3: I find it difficult to see major differences between panels A and B (also confounded by the difference in length of the two constructs). I would not conclude anything about significant changes from the presented data. Also, why is Tyr39 singled out? Is that the supposed contact point for Cys90 MTSL? (It is not particularly broadened).

A3: We have made the vertical scales in the panels A and B the same. FL- α -syn broadens in the N-terminal region and in residues around spin label due to the

intramolecular long-range interactions, which is weak for CT- α -syn, indicating a more open conformation. Tyr39 is singled out because its I(para)/I(dia) value is lower than that of other residues in the N-terminus, indicating that Tyr39 is close to MTSL, rather than a supposed contact point for Cys90 MTSL.

Q4. *It is very odd that the authors introduce new results in the Discussion (Fig. S4A-C). This should be included in the appropriate place in Results and be integrated into the narrative there.*

A4: We have integrated Figure S4 related results into Results.

Q5. *PDI seems an appropriate chaperone to investigate despite the lack of Cys residues (and disulfide bonds) in α -syn; however, additional references could be provided eg. PMID 33516852 and 32850841. However, it is not clear if the conclusions are general. The study would benefit greatly from inclusion of at least one other chaperone, e.g. Hsp70, which is considered a central chaperone in the whole PD field. Simply collecting data corresponding to Fig. 2 (at the very least at one Hsp70 concentration, but preferably more to gauge changes in affinity) would be a major improvement.*

A5: We now include PMID 33516852 and 32850841 as refs 53 and 61. We performed titration experiments using *Escherichia coli* DnaK (HSP70) (Figure R5). DnaK was titrated into 0.2 mM FL or CT- α -syn in the presence of 5 mM ADP and 5 mM MgCl₂, from mole ratios of 1:0.5 to 1:5. DnaK does not induce significant chemical shift perturbations in FL and CT- α -syn. Segments 1 and 2 of FL and CT- α -syn, however, experience signal attenuation upon adding DnaK, which suggests interactions. At a stoichiometry of 1:5 α -syn: DnaK, segment 1 of CT- α -syn shows stronger signal reduction than that of FL- α -syn, whereas segment 2 signals are almost equally attenuated, indicating stronger binding of segment 1 to DnaK caused by the C-terminal truncation. These results also reveal that the binding pattern of α -syn to DnaK is essentially identical to that of PDI, despite the difference in binding strength.

Figure R5. NMR titration of FL and CT- α -syn with DnaK. (A) Overlaid ^1H - ^{15}N HSQC spectra of 0.2 mM ^{15}N -enriched FL- (left) and CT- α -syn (right) in the absence (black) and presence (red) of 1.0 mM DnaK. (B) Residue-resolved signal attenuation (I/I_0) of FL- (black) and CT- α -syn (red) upon adding five equivalents of DnaK. Binding regions are colored gray. (C) Representative ^1H - ^{15}N HSQC cross-peaks within segments 1 and 2 of FL- (top) and CT- α -syn (bottom) at two titration points. (D) Statistical analysis of signal attenuation of FL- (black) and CT- α -syn (red) in the presence of five equivalents of DnaK.

Minor issues:

Q6. The authors should state clearly in Introduction/Results that CT- α -syn contains residues 1-99. This information is currently buried in the M&M section.

A6: We now state this in Introduction/Results (Lines 100 and 121).

Q7. Minor linguistic issues: “interfering the interaction”, “inhibition on fibrillation”, “C-terminal truncated”, “50% of FL- α -syn is bound state”

A7: We have corrected these problems.

Q8. There is nothing wrong with the sentence “These observations raise the question of whether C-terminal truncations affect the interaction of the N-terminal domain with membranes and chaperones, which could affect its functionality” but given the conclusions

from the manuscript and the preceding statement about C- and N-terminal interactions, I would suggest to rephrase to e.g. “These observations raise the question of whether C-terminal truncations free up the N-terminal domain to interact [more extensively?] with membranes and chaperones, thus affecting α -syn functionality”.

A8: We agree and have made this change.

Q9. Fig. S1A only differs from Fig. 1 in having a slightly different lipid composition (30% CL rather than 10%). This should be emphasized more clearly in the beginning of the figure legend to both Fig. 1 and Fig. S1. In that context “Data for CT- α -syn show that signals are more attenuated in the presence of liposomes containing CL” should be clarified to indicate that it is 30 versus 10% (there are no data for 0% CL).

A9: We have removed data for 10% CL. We now provide all the lipid compositions in Figure legend. To reveal the stronger membrane binding of CT- α -syn, we have rewritten the sentence to read “Data for CT- α -syn show that signals are more attenuated in the presence of liposomes containing CL, POPA or POPS compared to FL- α -syn,”

Q10. Fig 4A: why is there such a significant green signal for control as well? Is apoptosis expected to be wide-spread even in normal cells or are the cells stressed because of growth conditions? (I ask out of ignorance of practical cell biology).

A10: It is normal to observe significant green signal for cultured cancer cell lines.

Q11. Fig 4C: Can these data be densitometrically quantified? Difficult to assess as is.

A11: It is difficult to quantify all the bands, because some are too weak.

Q12. Fig. 4D: The blank controls are almost impossible to see (which of course emphasizes its similarity with FL- α -syn but should still be redrawn).

A12: We have redrawn Fig. 4D with the blank shown in purple.

Q13. There are no references to Figs. S2 and S3 in the text. The conclusion in Fig. S2 about a more extended conformation for CT α -syn needs to be elaborated (it is difficult for the untrained eye to see if the difference to FL α -syn is significant).

A13: As described above, we have removed 10% CL data so Figure S3 data now is placed in Figure 4.

The transverse relaxation rate enhancements (R₂) in Figure S2 are calculated with Eq:

$$R_2 = \frac{1}{T_b - T_a} \ln \frac{I_{dia}(T_b)I_{para}(T_a)}{I_{dia}(T_a)I_{para}(T_b)}$$

Figure S2 has been removed. All the supporting figures have been updated and indicated in appropriate places in the manuscript.

We hope our revised manuscript is satisfactory. Please contact me if you need more information.

Sincerely,

Conggang Li

Reviewers' comments:

Reviewer #1 (Remarks to the Author):

Accept

Reviewer #2 (Remarks to the Author):

The authors have satisfactorily answered Q1, Q2, and Q3 by providing explanations and additional experiments. However, I am not convinced by their explanation for Q4.

I am happy to see the size exclusion data performed by the authors but unfortunately, it confirms my alternative line of thought that CT- α -syn could aggregate into oligomeric intermediates in presence of SK-N-SH cells and could collapse the membrane potential and caused cyto c release. I am not convinced by the author's rationale as explained subsequently. a) The authors incubated 50 μ M CT- α -syn with SK-N-SH cells for 5 hours and observe a bimodal peak in size exclusion chromatography for the monomeric fraction (\sim 15.5 mL) and another peak with identical A280 absorbances that the authors rationalize as "the second peak may arise from the leaked protein due to toxicity of the CT- α -syn". I presume the authors mean leaked soluble proteins from the mitochondrial matrix or the SK-N-SH cytoplasmic proteins? A positive control is missing here showing that leaked proteins from mitochondria indeed elute at \sim 13 mL using a membrane permeabilizer. Alternatively, the authors should show a positive control for their antibody in the blot experiment and show that the used antibody, can detect aggregates of CT- α -syn. I am not sure which antibody was used for the immunoblot experiments in Figure R4. While most α -syn antibodies are directed towards the C-terminus, a positive blot for the second peak near 15.5 mL in the size exclusion data suggests the authors have used an antibody that binds N-terminal/NAC regions in the CT- α -syn. Could it not be possible that these regions could be shielded in the CT- α -syn oligomers preventing detection in an immuno-blot?

I strongly believe that the authors need to show that aggregation of CT- α -syn has not occurred during the time frame of their experiment with SK-N-SH cells. They have shown that no aggregation CT- α -syn occurs under static conditions but these experiments were carried out in absence of membranes or SK-N-SH cells. I understand aggregation experiments with SK-N-SH cells maybe not be possible with vigorous shaking environments required for α -syn aggregation. But membranes are well-known to promote aggregation of α -syn. The authors have a good set of results but they have to exclude that the mitochondrial membrane damage is not arising from oligomeric/aggregates of CT- α -syn.

Reviewer #3 (Remarks to the Author):

The authors have adequately addressed my concerns.

Dear Reviewer,

We appreciate your kind comments on our manuscript. We have tried to perform additional experiments to double check our results based on your suggestion. The specific response is as follows.

Q. The authors have satisfactorily answered Q1, Q2, and Q3 by providing explanations and additional experiments. However, I am not convinced by their explanation for Q4.

I am happy to see the size exclusion data performed by the authors but unfortunately, it confirms my alternative line of thought that CT- α -syn could aggregate into oligomeric intermediates in presence of SK-N-SH cells and could collapse the membrane potential and caused cyto c release. I am not convinced by the author's rationale as explained subsequently. a) The authors incubated 50 μ M CT- α -syn with SK-N-SH cells for 5 hours and observe a bimodal peak in size exclusion chromatography for the monomeric fraction (~15.5 mL) and another peak with identical A280 absorbances that the authors rationalize as "the second peak may arise from the leaked protein due to toxicity of the CT- α -syn". I presume the authors mean leaked soluble proteins from the mitochondrial matrix or the SK-N-SH cytoplasmic proteins? A positive control is missing here showing that leaked proteins from mitochondria indeed elute at ~13 mL using a membrane permeabilizer. Alternatively, the authors should show a positive control for their antibody in the blot experiment and show that the used antibody can detect aggregates of CT- α -syn. I am not sure which antibody was used for the immune-blot experiments in Figure R4. While most α -syn antibodies are directed towards the C-terminus, a positive blot for the second peak near 15.5 mL in the size exclusion data suggests the authors have used an antibody that binds N-terminal/NAC regions in the CT- α -syn. Could it not be possible that these regions could be shielded in the CT- α -syn oligomers preventing detection in an immuno-blot?

I strongly believe that the authors need to show that aggregation of CT- α -syn has not occurred during the time frame of their experiment with SK-N-SH cells. They have shown that no aggregation CT- α -syn occurs under static conditions but these experiments were carried out in absence of membranes or SK-N-SH cells. I understand aggregation experiments with SK-N-SH cells maybe not be possible with vigorous shaking environments required for α -syn aggregation. But membranes are well-known to promote aggregation of α -syn. The authors have a good set of results but they have to exclude that the mitochondrial membrane damage is not arising from oligomeric/aggregates of CT- α -syn.

A: We previously observed a bimodal peak in SEC after CT- α -syn being incubated with SK-N-SH cells for 5 h, with one corresponding to monomeric CT- α -syn and the other possibly being leaked proteins from cells. Based on your suggestion, we firstly used 0.5% Triton to destabilize membrane permeability of SK-N-SH cells (used as positive control) (Maria Célia Jamur, Constance Oliver. 2010, Methods Mol. Biol. 588:63-6; Stephanie Ghio et al. 2019, ACS. Chem. Neurosci. 10(8): 3815-3829). In this case, released intracellular substances eluted at ~13mL (Figure A), matching with the elution volume of leaked proteins caused by CT- α -syn. Next, to exclude the possibility that CT- α -syn oligomers may not be probed by the antibody (Abcam, ab 51252) that targets the N-terminal region

of α -syn (residue 1-100), we prepared CT- α -syn oligomers according to previous studies (Serene W Chen et al. 2015, Proc. Natl. Acad. Sci. USA. 112(16): e1994-2003; Giuliana Fusco et al. 2017, Science. 358(6369): 1440-1443). Briefly, lyophilized CT- α -syn was dissolved in PBS buffer (pH7.4) to a final concentration of \sim 800 μ M and incubated in 37 °C for 22 h without agitation. Fibrils formed during that time were removed by centrifugation at 15,000 g for 15 min. CT- α -syn monomers (10 kDa) and small oligomers were removed by filtration through 30 kDa cutoff membranes. Enriched oligomers were assessed by SEC, CT- α -syn monomers and oligomeric species eluted at \sim 15mL and \sim 5mL, respectively (Figure B). We then carried out a dot blot analysis to probe CT- α -syn oligomers. The result showed that this antibody (Abcam, ab 51252) targeting α -syn's N-terminus indeed detected CT- α -syn oligomers, consistent with the finding that the N-terminal region in α -syn oligomers is accessible (Giuliana Fusco et al. 2017, Science. 358(6369): 1440-1443).

We also used detergent-solubility assay to probe whether 50 μ M FL- or CT- α -syn formed fibrils in the presence of SK-N-SH cells for 5 h incubation (Klucken J, et al. 2004, J. Biol. Chem. 279(24): 25497-502; Dutta D, et al. 2021, Nat. Commun. 12(1): 5382). The immunoblot assay shows that α -syn only can be detected in the Triton X-100 soluble fraction, suggesting no fibril formation (Figure C). Taken together, our additional experiments confirm that CT- α -syn does not form detectable oligomers and fibrils during 5 h incubation with SK-N-SH cells, and the toxicity of CT- α -syn is mainly caused by enhanced monomer-membrane interactions.

Figure. (A) Size-exclusion chromatography of Triton-treated SK-N-SH cells (green) and CT- α -syn incubated with SK-N-SH cells for 5 h (red). (B) Size-exclusion chromatography and dot blot analysis of CT- α -syn oligomers and monomers. (C) Immunoblots of detergent-soluble and insoluble α -syn after 5 h incubation with cells. (Samples prepared as described by Klucken J, et al. 2004, J. Biol. Chem. 279(24): 25497-502; Dutta D, et al. 2021, Nat. Commun. 12(1): 5382). After 5 h incubation, Triton X-100 is added to FL- or CT- α -syn to a final concentration of 1% and incubated on ice for 30 min followed by centrifugation at 15000g for 15 min at 4 °C. The supernatant is referred to as the soluble fraction. The pellet is then resuspended in loading buffer (2% SDS), washed with PBS, and boiled for 15 min (insoluble fraction.)

We hope our response is satisfactory.

Sincerely,

Conggang Li

Conggang Li

REVIEWERS' COMMENTS:

Reviewer #2 (Remarks to the Author):

Dear Authors,

I am satisfied with the explanations and comprehensive additional experiments you have performed. I am convinced that the claims made are backed up sufficiently with data. I strongly recommend that the additional experiments you have performed be included in the supporting information (or as you see fit) since it includes important controls in my opinion. I appreciate and thank you for your patience with the review process.

Regards,

Reviewer 2